# RETURN AUGMENTATION GIVES SUPERVISED RL TEMPORAL COMPOSITIONALITY

## ABSTRACT

Offline Reinforcement Learning (RL) methods that use supervised learning or sequence modeling (e.g., Chen et al. (2021a)) work by training a return-conditioned policy. A fundamental limitation of these approaches, as compared to value-based methods, is that they have trouble generalizing to behaviors that have a higher return than what was seen at training (Emmons et al., 2021). Value-based offline-RL algorithms like CQL use bootstrapping to combine training data from multiple trajectories to learn strong behaviors from sub-optimal data. We set out to endow RL via Supervised Learning (RvS) methods with this form of temporal compositionality. To do this, we introduce SUPERB, a dynamic programming algorithm for data augmentation that augments the returns in the offline dataset by combining rewards from intersecting trajectories. We show theoretically that SUPERB can improve sample complexity and enable RvS to find optimal policies in cases where it previously fell behind the performance of value-based methods. Empirically, we find that SUPERB improves the performance of RvS in several offline RL environments, surpassing the prior state-of-the-art RvS agents in AntMaze by orders of magnitude and offering performance competitive with value-based algorithms on the D4RL-gym tasks (Fu et al., 2020).

## 1 INTRODUCTION

The use of prior experiences to inform decision making is critical to our human ability to quickly adapt to new tasks. To build intelligent agents that match these capabilities, it is natural to seek algorithms that learn to act from preexisting datasets of experience. Research on this problem, formally known as offline reinforcement learning (RL), has focused on two main approaches. The first takes existing off-policy RL algorithms, such as those based on Q-learning, and alters them to reduce issues caused by distributional shift. The resulting algorithms use value pessimism and policy constraints to keep actions within the support of the data distribution while simultaneously optimizing for high returns (Fujimoto et al., 2019; Kumar et al., 2020a). The second, RL via Supervised Learning (RvS), draws inspiration from generative modeling and supervised learning to learn outcome-conditioned policy models and uses them to predict which actions should be taken in order to get a high return (Schmidhuber, 2019; Chen et al., 2021b; Emmons et al., 2021).

RvS algorithms are appealing due to their simple training objective, robustness to hyperparameters, and strong performance, especially when trained in a multi-task setting. Recently, however, attention has been brought to their suboptimality in certain settings, such as stochastic environments (Paster et al., 2022; Villaflor et al., 2022; Eysenbach et al., 2022) and, as we remark in this work, offline settings where temporal compositionality[1] is required for good performance (see, e.g., Figure 1). Because return-conditioned RvS agents are trained using returns calculated from an offline dataset, they may fail to extrapolate to higher returns not present in any single trajectory but made possible by combining behaviors from multiple trajectories. While value-based methods use dynamic programming to compose behaviors across trajectories, RvS approaches fail to fully exploit the temporal structure inherent to sequential decision making (Sutton, 1988).

In our work, we aim to endow RvS methods with this kind of temporal compositionality. Traditionally, RvS agents are trained using return labels computed by summing over the future rewards of

---

[1]We define temporal compositionality as the composition of behaviors from different timesteps within a trajectory or from different trajectories.

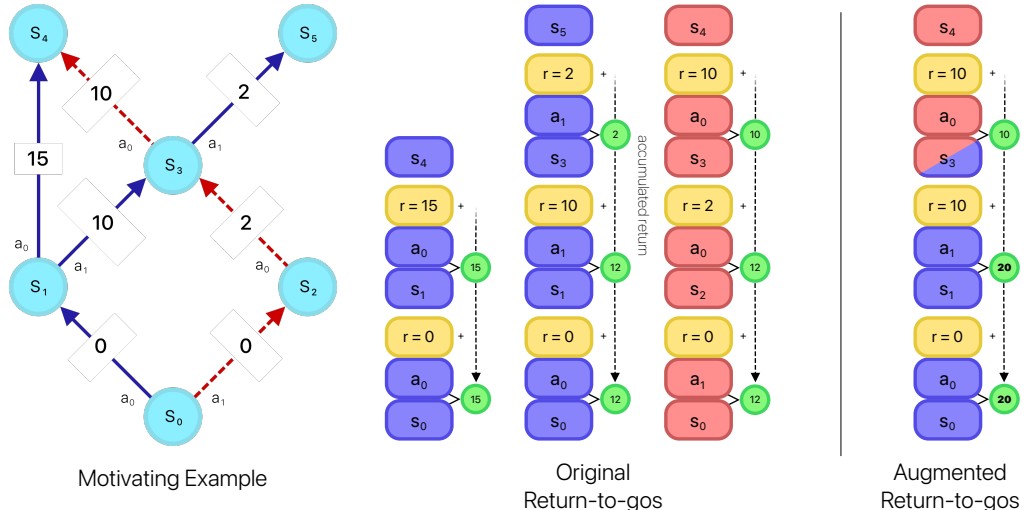

Figure 1: **Left:** In this illustrative offline RL task, the dataset consists of trajectories gathered by two suboptimal policies: the blue/solid policy, which goes to the left and either gets a return of 12 or 15, and the red/dashed policy, which goes to the right and gets a return of 12. **Middle:** Typical RvS algorithms compute the return-to-go by summing up the rewards along each empirical trajectory. While the optimal trajectory $(s_0, s_1, s_3, s_4)$ has a value of 20, no empirical trajectory has a value of 20. **Right:** Our method, SUPERB, combines rewards from multiple trajectories when calculating return-to-go labels and applies a label of 20 to each state and action along the optimal trajectory.

each trajectory individually. Our main insight is that we can apply the $n$-step temporal difference (TD) relation (Sutton & Barto, 2018) to offline data to augment the return labels used in training with returns that are made possible by composing different trajectories. While Q-learning algorithms aim to learn the value of the optimal policy, our method uses a distributional value function to assist in generating a distribution of feasible return labels on which we can train a return-conditioned RvS agent. We propose to iteratively apply our method to augment returns, which we show can generate exponentially more labels for RvS agents to train on. Our main contributions are:

1. We show that there are some environments where, both analytically and empirically, a return-conditioned RvS agent falls behind value-based agents that use bootstrapping, and propose a data augmentation method that bridges this gap by using the $n$-step TD relation to combine rewards from different trajectories.

2. We show that temporal compositionality can exponentially improve the number of training trajectories and is necessary to learn optimal policies in some offline RL datasets.

3. We evaluate our method on the D4RL offline RL suite. Our method dramatically improves the performance of RvS on AntMaze environments, where optimal policies must stitch together behaviors from partial demonstrations, from near zero to state-of-the-art. We also obtain performance competitive with value-based methods on the D4RL-gym tasks.

## 2 METHOD

### 2.1 PRELIMINARIES

We consider the offline Reinforcement Learning (RL) setting where the environment is modeled as a Markov Decision Process (MDP), $\mathcal{M} = \langle \mathcal{S}, \mathcal{A}, T, R, \gamma \rangle$, consisting of a state space, action space, transition function, reward function, and discount factor, respectively (Lange et al., 2012; Sutton & Barto, 2018). The agent is given fixed dataset $\mathcal{D}$ of state-action-reward trajectories $\{\tau = (s_1, a_1, r_1, s_2, a_2, r_2, \dots)\}$ produced by a (non-Markovian) "empirical policy" $\pi_e$ acting in $\mathcal{M}$, and tasked with learning a policy $\pi_\theta : \mathcal{S} \to \mathcal{A}$ that obtains high return $\sum_t \gamma^t r_t$ when executed in $\mathcal{M}$.

We build on the Reinforcement Learning via Supervised Learning (RvS) paradigm (Chen et al., 2021b; Schmidhuber, 2019), in which the agent learns a return-conditioned policy $p_\theta(a \mid s, z)$, where $z \in \mathcal{Z}$ is an empirical statistic about trajectories that start in state $(s, a)$, and then conditions this policy on a favorable $z^*$ to obtain its policy $\pi_\theta(a \mid s) = p_\theta(a \mid s, z^*)$. We condition on the original discounted return objective $z = g = \sum_t \gamma^t r_t$, and leave extensions to other potential $\mathcal{Z}$ to future work. We define a return function $G : \tau \mapsto \mathbb{R}$ that maps from (sub)trajectories to their discounted return. $p_\theta$ is trained to model empirical samples $(s, a, g) \sim \mathcal{D}$ using maximum likelihood estimation, where $(s, a)$ are sampled uniformly from all $(s, a)$ pairs, and $g = G(\tau_{sa})$ is the return of the empirical subtrajectory that begins with $(s, a)$. As in prior works (Chen et al., 2021a; Emmons et al., 2021), $z^*$ is tuned along with other hyperparameters to maximize performance.

## 2.2 BOOTSTRAPPING

Many of the most successful RL methods such as Q-Learning (Watkins & Dayan, 1992) exploit the time structure of MDPs to do temporal difference learning (Sutton, 1988; Tesauro, 1995), a form of dynamic programming where a learned model of the expected return is used to provide learning targets, either recursively for itself or for other agent modules. This process, which has come to be known as "bootstrapping", allows the agent to learn off-policy and achieve higher returns than those observed by the empirical policy. One way to understand bootstrapping is that it allows the agent to implicitly stitch together empirical experiences via the learned return model (Sutton, 1988). This effect is particularly visible in the offline RL setting, where Q-learning based methods such as BCQ (Fujimoto et al., 2019) and CQL (Kumar et al., 2020a) are able to far surpass behavior cloning (BC) and obtain higher returns than the empirical policy $\pi_e$.

Unlike Q-learning based methods, the ordinary approach to training $p_\theta$ in RvS methods does not use bootstrapping: $g$ is simply computed as the cumulative (discounted) reward of the trajectory starting in $(s, a)$. For this reason, the performance of RvS methods is fundamentally limited by the return of the best performing trajectories in $\mathcal{D}$, since these upper bound the set of returns $\{G(\tau_i)\}$ used for training $p_\theta$. While extrapolation via function approximation is possible and conditioning on the highest observed returns allows RvS agents to outperform the *average* empirical policy, typical RvS training ignores the known temporal structure of $\mathcal{M}$ and lags behind bootstrapping methods on certain datasets (Emmons et al., 2021).

On other datasets, however, RvS methods still outperform Q-learning based methods and are known to be more stable when using varied hyperparameters (Emmons et al., 2021; Ghosh et al., 2021). One potential reason for this is that (1) the self-supervised bootstrapping used by Q-learning, when combined with (2) function approximation and (3) off-policy data, suffers from a general source of training instability known as the "deadly triad" (Sutton & Barto, 2018). By contrast, RvS methods use (by assumption) unbiased return labels generated by Monte Carlo rollouts of $\pi_e$.

## 2.3 SUPERB: SUPERVISED RL WITH BOOTSTRAPPING

To resolve a fundamental limitation of RvS methods and grant them the ability to surpass the highest returns in the empirical dataset, we propose to augment the empirical data $\mathcal{D}$ with "bootstrapped" returns calculated by combining multiple trajectories. Recall that in return-conditioned RvS, training data consists of $(s, a, g)$ triples, where $g$ is a possible return that can result from taking action $a$ in state $s$. We augment the dataset with novel possible returns by observing that the return $G(\tau_1 \tau_2)$ of two concatenated subtrajectories can be computed from parts using the $n$-step TD relation:

$$G(\tau_1 \tau_2) = G(\tau_1) + \gamma^{|\tau_1|} G(\tau_2) \tag{1}$$

To generate returns that could occur by stitching two subtrajectories together, we model the distribution of the returns of trajectories that start at the last state of $\tau_1$, and create augmented returns by replacing $G(\tau_2)$ in Equation 1 with samples from this distribution. Our method, which we call SUPERB, for Supervised RL with Bootstrapping, consists of the following phases:

**Learning a Model of Returns.** The input to our algorithm is a dataset of states, actions, and returns $(s, a, g) \sim \mathcal{D}$. As the first step towards augmenting the returns, we learn a model $V_\phi(G \mid s)$ of the distribution of returns possible from any given state in the dataset, parameterized by $\phi$.

**Augmenting the Dataset with New Returns.** In the second phase of our method, we strive to create novel return labels that represent possible values an agent could achieve from each $(s, a)$ pair

**Algorithm 1** SUPERB

---

**function** SUPERB(data $\mathcal{D} = \{\tau\}$, steps $N$)
    $\mathcal{D}^0 = \mathcal{D}$
    **for** $n = 1 \dots N$ **do**
        $V_\phi^{n-1} \leftarrow$ TRAINREWARDMODEL($\mathcal{D}^{n-1}$)
        $\mathcal{D}^n =$ AUGMENTDATA($\mathcal{D}^{n-1}, V_\phi^{n-1}$)
    $p_\theta =$ TRAINRVSAGENT($\mathcal{D}^N$)
    **return** $p_\theta$

**function** TRAINREWARDMODEL($\mathcal{D} = \{s, g\}$)
    Sample $(s, g) \sim \mathcal{D}$
    Train $V_\phi(G \mid s)$ using MLE
    **return** $V_\phi$

**function** AUGMENTDATA($\mathcal{D} = \{\tau\}$, model $V_\phi$)
    $\mathcal{D}' = \{\}$
    **for** $m = 1 \dots M$ **do**
        Sample $\tau \sim \mathcal{D}$
        Choose prefix $\tau_1 = (s, a, r, \dots s')$ of $\tau$
        $g' = G(\tau_1) + \gamma^{|\tau_1|}g, \quad g \sim_w V_\phi(s')$
        $\mathcal{D}' \leftarrow (s, a, g')$
    **return** COMBINE($\mathcal{D}, \mathcal{D}'$)

**function** TRAINRVSAGENT($\mathcal{D} = \{(s, a, g)\}$)
    Sample $(s, a, g) \sim \mathcal{D}$
    Train $p_\theta(a \mid s, g)$ using MLE
    **return** $p_\theta$

---

in the dataset. To do this, SUPERB chooses a prefix for each trajectory in the dataset, denoted $\tau_1 = (s, a, r, \dots, s')$, and combines it with another subtrajectory $\tilde{\tau}$ in the dataset to imagine a hypothetical new trajectory. Rather than explicitly combine trajectories and calculate returns, SUPERB uses the learned return model. Specifically, rather than calculate return associated with $(s, a) \sim \mathcal{D}$ as $G(\tau_{sa}) = \sum_t \gamma^t r_t$ as in prior return-conditioned RvS works (Chen et al., 2021a; Emmons et al., 2021), SUPERB uses the $n$-step TD relation in Equation 1 to calculate the return of an imagined, combined trajectory as $G(\tau_1 \tilde{\tau}) = G(\tau_1) + \gamma^{|\tau_1|}g_{s'}$, where $g_{s'}$, sampled from the learned return model $V_\phi(G \mid s')$, represents the value of a possible subtrajectory that starts in $s'$. These returns are added to the new, augmented dataset $\mathcal{D}'$.

Performing these two operations only once gives us returns that are stitched from at most two subtrajectories. To generate labels that compose $2^N$ subtrajectories, we repeat this process $N$ times.

**Training the Return-Conditioned Policy.** Finally, a return-conditioned RvS policy $p_\theta(a \mid s, g)$ is learned via maximum likelihood from $\mathcal{D}'$ as done in prior works (Chen et al., 2021a; Emmons et al., 2021; Schmidhuber, 2019; Kumar et al., 2019). The full process is summarized in Algorithm 1.

## 2.4 BIASING SUPERB TOWARDS RETURN MAXIMIZATION

The implementation of the different phases in SUPERB involves several important design choices: what distributional return model to use for $V_\phi(G \mid s)$, what distribution over possible $\tau_1$ to use when forming the bootstrapped returns, how to sample from $V_\phi(G \mid s)$, and how much of the original dataset $\mathcal{D}$ to keep. We opt to inform our choices based on the downstream use of the agent: return maximization. In particular, since the RvS policy typically conditions on high returns, we make several choices to maximize the performance of the agent in this regime.

First, rather than sample directly from $V_\phi(G \mid s)$ when augmenting returns, which just as likely to form low returns as it is high returns, we learn $V_\phi(G \mid s)$ using quantile regression as in Dabney et al. (2018) and bias our samples by only drawing from the upper quantiles (specifically, we use the mean of the top 5 of 20 quantiles). Second, when choosing $\tau_1$, which represents the first part of a trajectory to be spliced together with a return from a different trajectory, we use a backward induction approach to consider all possible $\tau_1$, only choosing to augment using $\tau_1$ when the augmented return is higher than the existing return. Finally, since as a consequence of this decision the augmented returns are guaranteed to be higher than the original returns, we choose to discard the returns in the original dataset when training the RvS policy. See Appendix B for further details.

These design choices introduce some similarity to Q-learning methods. However, while Q-learning aims to find a Q-function for the optimal policy by minimizing the Bellman error, the objective of SUPERB is to alter and expand the distribution of returns on which an RvS agent is trained. Despite these different motivations, biasing SUPERB toward return maximization allows it to be understood as way of unifying RvS and Q-learning (cf. TD($\lambda$), Sutton (1988)): as the number of bootstrap iterations increases, the augmented dataset produced by SUPERB becomes increasingly similar to a sample-based representation of an optimal Q function. By using an intermediate number of bootstrap iterations, we hypothesize that we can trade-off the strengths of RvS and Q-learning (i.e., training

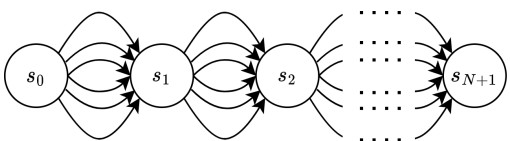

Figure 2: **$N$-Step Chain MDP:** The MDP is initialized in $s_0$ and terminates in $s_{N+1}$, with actions always leading to the next state in the chain. In this MDP there are $b^N$ possible trajectories.

stability in case of no bootstrapping vs. the ability to stitch together a large number of behaviors). Our intermediate approach may be helpful in many realistic offline RL datasets (such as those found in D4RL (Fu et al., 2020)), which only require a small amount of stitching to find strong behaviors.

### 2.5 ANALYSIS

In this section, we show formally how temporal composition is necessary to generate optimal reward labels for some sets of offline RL problems, as well as how it can generally improve the sample complexity of return-conditioned RvS methods.

**Proposition 1** (Necessity). *For any positive integer $n$, there exists an MDP $\mathcal{M}$ and data generating policy $\pi_e$ for which $n$-step bootstrapping is strictly necessary to generate dataset $\mathcal{D}$ containing $(s, a, g^*)$, where $g^*$ is the maximum reward achievable for a trajectory starting in $(s, a)$.*

*Proof.* See Figure 1 for the case $n = 1$. For the general case, one can construct an MDP and offline dataset where non-Markov $\pi_e$ is generated by $N + 1$ different policies, each containing a segment of the optimal policy. The details are found after the proof of Proposition 2a in Appendix A. □

Note that the proof of Proposition 1 explicitly constructs a non-Markovian $\pi_e$. Some thought reveals that non-Markovian $\pi_e$ is required for necessity: if we continually sample data from Markovian $\pi_e$, we will eventually sample the highest possible return that could be reached by augmenting returns produced by $\pi_e$. This suggests that we should expect SUPERB to provide more benefit when augmenting datasets produced by non-Markovian policies $\pi_e$, such as is the case when the dataset is produced by a mixture of policies.

The following Proposition 2 (and Proposition 2A in Appendix A) assumes an "$N$-Step Chain MDP", exemplified in Figure 2, with $N + 1$ states, and $b$ actions in each state. The Chain MDP simplifies analysis in a way that is particularly favorable toward bootstrapping. Although the results will not hold generally, they demonstrate that bootstrapping has the potential to be extremely valuable in environments where trajectories have many possible points of overlap. Indeed, in the Chain MDP, *every* trajectory overlaps with every other trajectory at every state. On the other end of the spectrum, we could consider an MDP where the only point of overlap is in the initial state: in such an MDP, there would be no advantage to bootstrapping.

**Proposition 2** (Coverage of Trajectory Space). *In a deterministic $N$-step Chain MDP with branching factor $b$, it is possible for $N$-step bootstrapping to (implicitly) capture full "coverage" of the size $b^N$ trajectory space with only $b$ empirical trajectories—an exponential increase in coverage relative to no bootstrapping. Coverage here refers to the percentage of unique trajectories present in the dataset, where we consider two trajectories with the same actions taken in every state as equivalent.*

*Proof.* We apply our Augmentation-Bootstrapping Equivalence Lemma, found in Appendix A, which states that return label bootstrapping is equivalent to augmenting the set of trajectories by stitching together subtrajectories. While the equivalence itself is interesting, a formal statement of the lemma is long and relegated to Appendix A. □

Our analysis suggests that SUPERB can offer a nontrivial advantage over plain RvS methods; however, the maximization bias we introduce via our design choices, together with the approximation error introduced by our model of the return distribution, suggests a tradeoff between the benefit of bootstrapping and risk of increasing bias. As a result, the right number of bootstrapping iterations may be problem dependent. In the next section, we explore how SUPERB performs empirically.

## 3 EXPERIMENTS

We designed our experiments to answer the following questions:

- Can SUPERB improve RvS performance on environments requiring temporal composition?
- How does performance vary with the number of bootstrapping iterations?
- What design choices are important when doing return augmentation?

Key experimental details are described in the main text, with minor details left to Appendix B.

### 3.1 BENCHMARK TASKS

We conduct our experiments on the D4RL benchmark suite (Fu et al., 2020), following precedent set by prior works in offline RL. In particular, we focus on the Gym MuJoCo tasks (Brockman et al., 2016), which include Hopper, HalfCheetah, and Walker2d, as well as the AntMaze tasks ("umaze", "medium", and "large" mazes). The gym tasks are designed to test how offline RL algorithms deal with suboptimal agents and narrow data distributions, while the AntMaze tasks test their ability to learn from sparse reward data generated by non-Markovian policies. We are primarily interested in performance on the AntMaze tasks, where strong performance requires stitching together behaviors from different trajectories, as the vast majority of the datasets consists of trajectories that do not start at the initial state and travel to the goal. As reported by Emmons et al. (2021), RvS methods perform poorly on these tasks without additional information (e.g., RvS-G in Emmons et al. (2021) does goal relabeling using the knowledge of the state dimensions that correspond with the goal state).

### 3.2 BASELINES

We base our implementation of RvS on the code provided by Emmons et al. (2021), and rerun the RvS baseline with any improvements we made to the codebase to ensure a fair comparison. We compare with behavioral cloning and filtered behavioral cloning (Chen et al., 2021a), as these have similar implementations to RvS and provide a simple sanity check. We also compare with Decision Transformer (Chen et al., 2021a), a state-of-the-art implementation of RvS which uses a transformer (Vaswani et al., 2017) as a policy. Finally, we compare with several Q-learning variants. We compare our method with CQL (Kumar et al., 2020b), which is the standard for many offline RL tasks, and IQL (Kostrikov et al., 2021), which has similarities with our method as discussed in Section 4. We report results for CQL, BC, %BC, DT, and RvS as reported in Emmons et al. (2021). For IQL (Kostrikov et al., 2021), we re-run their AntMaze experiments on the updated v2 version of the environments and use their originally reported results otherwise. To ensure fair comparison, we rerun RvS using our own hyperparameters and report the results under the name RvS (Ours).

### 3.3 PERFORMANCE ON D4RL TASKS

Table 4 shows a comparison of the performance of our method with prior methods. Our experiments reveal that SUPERB outperforms the base RvS agent across almost all tasks, especially on the medium and large AntMaze environments, where the augmented returns generated by SUPERB boost performance to be state-of-the-art. In contrast with prior works (e.g., Emmons et al. (2021); Janner et al. (2021)), which use goal-state information to achieve high performance in AntMaze, SUPERB achieves strong performance only using basic information from the reward signal. On the D4RL gym tasks where temporal compositionality is less important, SUPERB generally retains the performance of RvS without any augmentation. Consistent with our hypothesis from section 2.5 that SUPERB provides the most benefit on datasets formed by non-Markovian $\pi_e$, a non-trivial improvement is obtained on two out of the three of the `medium-replay` datasets, which are the only non-Markovian gym datasets where expert performance is not obtained.

Our results are more modest on the other D4RL-Gym tasks, where the datasets are single-task and, in many cases, only collected with a single Markov policy. We believe, however, that most real-world offline-RL datasets are likely to be multi-task and gathered by a mixture of different policies, where SUPERB will have a large effect.

| Task | CQL | IQL | BC | % BC | DT | RvS-R | RvS (Ours) | SUPERB |
|---|---|---|---|---|---|---|---|---|
| antmaze-umaze-v2 | 44.8 | 83.7 | 54.6 | 60.0 | 65.6 | 64.4 | 84.2 | **85.4** |
| antmaze-umaze-diverse-v2 | 23.4 | 66.7 | 45.6 | 46.5 | 51.2 | 70.1 | **77.7** | 76.9 |
| antmaze-medium-play-v2 | 0.0 | 74.7 | 0.0 | 42.1 | 1.0 | 4.5 | 2.7 | **76.5** |
| antmaze-medium-diverse-v2 | 0.0 | 71.0 | 0.0 | 37.2 | 0.6 | 7.7 | 2.7 | **78.1** |
| antmaze-large-play-v2 | 0.0 | **44.3** | 0.0 | 28.0 | 0.0 | 3.5 | 0.8 | 25.8 |
| antmaze-large-diverse-v2 | 0.0 | **49.0** | 0.0 | 34.3 | 0.2 | 3.7 | 6.9 | 36.9 |
| halfcheetah-random-v2 | **18.6** | - | 2.3 | 2.0 | 2.2 | 3.9 | 2.2 | 2.2 |
| hopper-random-v2 | 9.3 | - | 4.8 | 4.1 | 7.5 | 7.7 | 5.8 | **14.3** |
| walker2d-random-v2 | 2.5 | - | 1.7 | 1.7 | 2.0 | -0.2 | **6.0** | 5.6 |
| halfcheetah-medium-replay-v2 | **47.3** | 44.2 | 36.6 | 40.6 | 36.6 | 38.0 | 39.1 | 39.1 |
| hopper-medium-replay-v2 | **97.8** | 94.7 | 18.1 | 75.9 | 82.7 | 73.5 | 77.9 | 90.3 |
| walker2d-medium-replay-v2 | **86.1** | 73.9 | 26.0 | 62.5 | 66.6 | 60.6 | 50.5 | 66.2 |
| halfcheetah-medium-v2 | **49.1** | 47.4 | 42.6 | 42.5 | 42.6 | 41.6 | 42.3 | 42.1 |
| hopper-medium-v2 | 64.6 | 66.3 | 52.9 | 56.9 | 67.6 | 60.2 | 58.9 | 56.8 |
| walker2d-medium-v2 | **82.9** | 78.3 | 75.3 | 75.0 | 74.0 | 71.7 | 72.4 | 74.7 |
| halfcheetah-medium-expert-v2 | 85.8 | 86.7 | 55.2 | **92.9** | 86.8 | 92.2 | 92.7 | 92.5 |
| hopper-medium-expert-v2 | 102.0 | 91.5 | 52.5 | **110.9** | 107.6 | 101.7 | 109.8 | 110.1 |
| walker2d-medium-expert-v2 | 109.5 | **109.6** | 107.5 | 109.0 | 108.1 | 106.0 | 106.9 | 107.9 |

Table 1: We evaluated our method with two bootstrap iterations on the AntMaze and Gym tasks from D4RL (Fu et al., 2020). Across the AntMaze tasks, which require stitching together behaviors from several suboptimal trajectories in order to perform well, SUPERB provides a strong performance improvement over the baseline RvS agents that don't use bootstrapping, outperforming the prior state-of-the-art in 3 of the 6 mazes. The benefits of SUPERB on the Gym tasks are more modest, though we observe improvements over RvS on a few of the suboptimal datasets, particularly those where the dataset was generated by a non-Markovian combination of policies. * **Bold** denotes the best overall method, underline denotes the best RvS method. Results are averaged over 5 seeds.

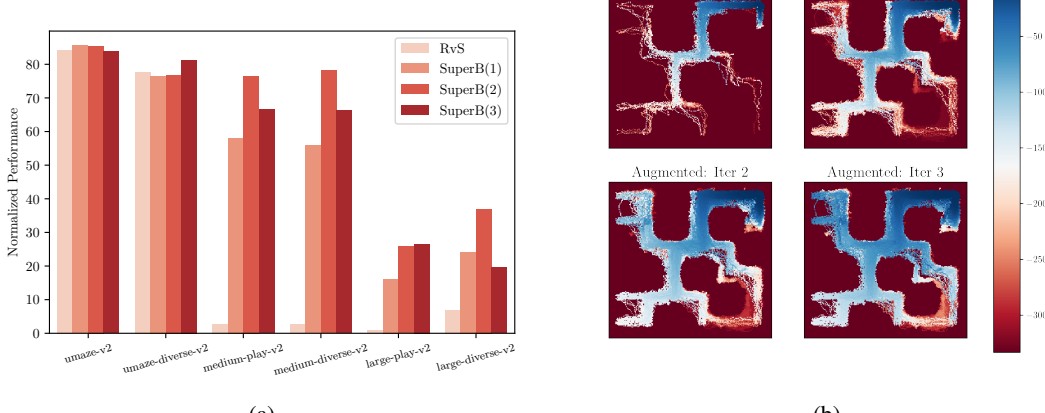

(a)                                                    (b)

Figure 3: **Left:** In the AntMaze tasks that include data generated by suboptimal agents, SU-PERB provides a strong benefit. After two iterations of SUPERB, performance starts to decrease, indicating that the benefit provided by the extra bootstrapping step is outweighed by the additional errors added from function approximation. **Right:** We plot the augmented returns in `antmaze-medium-diverse-v2` for up to three iterations in order to visualize the effect of SU-PERB on the maximum values associated with the trajectories in the dataset. Each pixel is colored according to the highest augmented return across observed states that fall within that pixel, defaulting to $V_{min}$ if no states were observed (possible returns are in $[V_{min}, V_{max}] = [-1/(1-0.997), 0]$). Even a single iteration serves to make the coverage of successful examples significantly more robust.

## 3.4 PERFORMANCE VS. BOOTSTRAP ITERATIONS

SUPERB can be applied iteratively $N$ times to create return labels that are bootstrapped $N$ times, equivalent to combining returns from up to $N + 1$ different trajectories. We ran SUPERB for up to three iterations on AntMaze in order to study the effect of this hyperparameter. Figure 3a shows that

| Ablation Study | SB(1) | SB(2) |
|---|---|---|
| SUPERB (Ours) | **55.8** | **78.1** |
| Remove Reward Transformation & Discounting | 7.1 | 3.8 |
| Remove Dynamic Target Return | 41.67 | 32.1 |
| Keep Original Returns from $\mathcal{D}$ | 23.7 | 22.1 |

Table 2: In our ablation study in `antmaze-medium-diverse-v2`, we find that a combination of a modified reward function (matching the implementation in the Q-learning based baselines), dynamic target returns, and reducing the dataset size by pruning the original return labels is necessary to achieve strong performance. Note that several of these ablations are still far stronger than the other strongest RvS method on this task, RvS-R, which only gets a score of 7.7.

tasks that benefit from bootstrapping (the medium and large AntMaze environments), most of the performance benefit is present after only 1-2 iterations. Figure 3b shows the maximum return label for each state in the `antmaze-medium-diverse-v2` task. Our visualization of the returns used to train prior RvS methods (such as RvS-R, DT, and SB(0) in Table 4) in Figure 3b (top left) shows that the dataset contains examples of successful trajectories, but that these trajectories are sparse. Just one or two iterations of SUPERB makes the coverage of successful trajectories far more robust, and we suspect further iterations only serve to add noise to the dataset. This empirical finding is reassuring since it suggests that in practice even one iteration of bootstrapping with SUPERB can provide a large benefit in practical datasets. However, as discussed in Section 2, one can imagine tasks more similar to the chain MDP which would benefit from many more iterations, meaning that there is no universally optimal number of iterations.

### 3.5 IMPORTANT DESIGN CHOICES

As discussed in section 2.4, there are several important choices when implementing SUPERB. Table 2 shows the result of our ablation study on the `antmaze-medium-diverse-v2` environment where we empirically verify that each of the following elements is important.

**Reward Transformation and Discounting in AntMaze.** In the AntMaze environment, it is common for value-based offline RL algorithms (such as IQL (Kostrikov et al., 2021)) to apply a transformation to the reward to convert it from a sparse reward of $1$ when the goal is reached to a reward of $0$ when a goal is reached and $-1$ each step otherwise. We found that applying this reward transformation combined with a high discounting factor of $0.997$ improved performance of SUPERB, but did not improve the performance of our RvS baseline.

**Dynamic Target Return Selection.** RvS policies need to be conditioned on a target return. In most prior work, this target is manually chosen and updated throughout the episode to account for rewards the agent receives. We take inspiration from Lee et al. (2022) and condition on the highest return quantile of our learned model $V_\phi(G \mid s)$ for the current state during the course of an episode. We found this improves performance in some tasks, as compared to manually choosing the target.

**Discarding Original Returns.** In traditional data augmentation methods, the original data is kept as a part of the dataset. However, we found discarding all but the most bootstrapped returns can improve performance dramatically, perhaps due to easier policy learning.

### 4 RELATED WORK

**Offline Reinforcement Learning** In the Offline RL setting, an agent must learn a policy that maximizes returns from a static dataset of trajectories (Lange et al., 2012). When doing RL with offline datasets, the goal is typically to compose behaviors seen in the offline dataset while avoiding taking out-of-distribution actions that may lead to unexpected outcomes (Levine et al., 2020). Offline RL algorithms typically address this with either value conservatism to penalize state-action pairs that don't have data coverage, policy constraints to keep the learned policy within the support of the dataset's distribution, or ensembles (Kumar et al., 2020b; Fujimoto et al., 2019; Agarwal et al., 2020). There are several standardized benchmark suites for offline RL, including D4RL (Fu et al., 2020) and RL Unplugged (Gülçehre et al., 2020), which include primarily deterministic tasks with

datasets generated by one or several suboptimal policies. Implicit Q-Learning (IQL) (Kostrikov et al., 2021) is similar to SUPERB in that it uses the upper range of a distributional value function and uses it to back out a policy without sampling or evaluating out-of-dataset actions. In contrast to IQL, SUPERB is motivated as a data augmentation scheme for RvS agents; further, it does not seek to minimize a 1-step Bellman error, approximates the distribution of potential returns rather than an upper expectile of such a distribution, and uses a different approach to policy learning.

**RL via Supervised Learning**  Our work builds off of the paradigm for RL first introduced in Schmidhuber (2019) and Kumar et al. (2019) where a predictive model is trained to predict actions conditioned on a desired outcome such as return. We refer to this class of algorithms as RvS, or RL via supervised learning, following Emmons et al. (2021). Decision Transformer uses a large transformer model that conditions on trajectory history and desired returns and is evaluated on offline RL tasks, demonstrating that RvS methods can be competitive in this domain (Chen et al., 2021a). Others have used RvS to learn goal-conditioned policies, such as in Ghosh et al. (2021) and Paster et al. (2021). While RvS methods fall behind other model-free algorithms in online RL, they are competitive in most popular offline RL benchmarks. In a recent work on tuning such methods (Emmons et al., 2021), return-conditioned RvS with a two-layer MLP was shown to be competitive with Decision Transformer and value-based methods on many tasks.

**TD-Learning**  Our proposed method takes advantage of the $n$-step TD relation (Sutton (1988), Chapter 7). Rather than use dynamic programming to learn a specific policy, however, we capture the data efficiencies of TD learning by explicitly augmenting a dataset of Monte Carlo returns (Sutton, 1988). Because the distributional value functions learned by SUPERB are initially grounded by supervised targets, this gives it a similar flavor to fixed or finite horizon TD methods (De Asis et al., 2020), which are resistant to the deadly triad, a common source of training instability for deep RL methods (Van Hasselt et al., 2018).

**Data Augmentation in RL**  Data augmentation has proven to be a powerful technique for improving RL sample efficiency in a variety of common scenarios, domain adaptation and transfer (Andrychowicz et al., 2020), RL from pixels (Yarats et al., 2020; Laskin et al., 2020), goal-driven RL (Kaelbling, 1993), and object-oriented RL (Pitis et al., 2020). In contrast with the aforementioned methods, which augment an agent's observations, our method does data augmentation on returns. This is similar to methods such as ESPER (Paster et al., 2022), which exchanges return-to-go with a learned return label to improve performance in stochastic environments, and RUDDER (Arjona-Medina et al., 2019), which learns a redistributed reward function to improve TD learning in environments with delayed rewards. When a simulator is available, human demonstration data may also be augmented "online" via imitation, as done by Mandlekar et al. (2020), who similar to SUPERB, use their imitation policy to stitch together segments from different demonstrations.

## 5 CONCLUSION

RvS approaches such as Decision Transformer have gained traction recently, but performance on tasks that require the agent to stitch together behaviors from different trajectories has lagged behind that of value-based methods (Chen et al., 2021a; Emmons et al., 2021). We quantify this phenomenon, giving examples of MDPs where the ability to compose behaviors from different trajectories gives an exponential improvement in trajectory coverage, or is outright required to solve the task optimally. To improve RvS, we proposed SUPERB, which augments the returns on which RvS agents are trained with returns computed using an $n$-step TD relation. On the D4RL benchmark suite, SUPERB dramatically improves the performance of RvS, achieving state-of-the-art performance on half of the AntMaze tasks and improving performance on almost all D4RL-gym tasks.

SUPERB provides a simple and compelling offline data augmentation method for an increasingly important class of RL algorithms. Conveniently, while return augmentation can be applied iteratively, even one or two iterations of data augmentation with SUPERB can dramatically improve performance. For future work, we plan to investigate further into how the number of data augmentation iterations can be tuned in an offline manner, as well as ways to apply these techniques to non-RL domains that use sequence modeling, such as language modeling. Future work might also consider scaling SUPERB to larger offline datasets in complex environments where coverage gaps are inevitable and data must collected from a wide range of different policies. Finally, it may be interesting to extend SUPERB to other empirical statistics that RvS methods might condition on.

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

# A  PROOF OF PROPOSITIONS

We prove Proposition 2 before Proposition 1. We use the following Lemma, which is stated in two parts for clarity (part (B) generalizes part (A)). The Lemma considers the "$N$-Step Chain MDP", exemplified in Figure 2, with $N+1$ states, and $b$ actions in each state, which each may have different reward distributions.

**Lemma 1** (Augmentation-Bootstrapping Equivalence)**.**

*(A)* *For any particular 2-step trajectory, $\tau = (s_0, a_0, r_0, s_1, a_1, r_1, s_2)$, in a 2-Step Chain MDP, where the subtrajectory $\tau_0 = (s_0, a_0, r_0, s_1)$ appears $n$ times in a given empirical dataset, and the subtrajectory $\tau_1 = (s_1, a_1, r_1, s_2)$ appears $m$ times, the following return estimators are equivalent:*

    *1. Augmentation:*
- *First form an augmented dataset of $nm$ trajectories, by concatenating each pair of trajectory segments $\tau_0$ and $\tau_1$. Then estimate the return of $\tau$ as the empirical average of its return in the augmented dataset:*

$$G(\tau) = \frac{\sum_{i=1}^{nm} r_0^i + r_1^i}{nm}$$

    *2. Bootstrapping:*
- *First estimate the return of $\tau_1$ as $G(\tau_1) = \sum_i^m r_1^i / m$. Then estimate the return of $\tau$ using the bootstrapped estimator:*

$$G(\tau) = \frac{\sum_{i=1}^{n} r_0^i + G(\tau_1)}{n}$$

*(B)* *Given an N-step Chain MDP and an N-step trajectory $\tau$, whose 1-step subtrajectories, $\{\tau_j = (s_j, a_j, r_j, s_{j+1})\}$, each appear $m_j$ times in the empirical dataset, the following return estimators are equivalent:*

    *1. Augmentation:*
- *First form an augmented dataset of $\Pi_j m_j$ trajectories, by concatenating each combination of trajectory segments. Then estimate the return of $\tau$ as the empirical average of its return in the augmented dataset:*

$$G(\tau) = \frac{\sum_{i=1}^{\Pi_j m_j} \sum_{k=0}^{N-1} r_k^i}{\Pi_j m_j}$$

    *2. $N-1$ steps of Bootstrapping:*
- *Denoting the trajectory slice from $s_k$ to $s_N$ as $\tau_{k:}$, estimate the return of $\tau$ using the bootstrapped estimator:*

$$G(\tau_{k:}) = \frac{\sum_{i=1}^{m_k} r_k^i + G(\tau_{k+1:})}{m_k}$$

    *where $G(\tau_{N:}) := 0$.*

    *3. Sum of local reward estimators:*
- *Estimate the return of $\tau$ as a sum of the returns of its 1-step components:*

$$G(\tau) = \sum_{k=1}^{N} \frac{\sum_{i=1}^{m_k} r_k^i}{m_k}$$

    *where $G(\tau_{N:}) := 0$.*

*Proof.* Part (A) is a special case of Part (B)(1)-(2). Both follow by induction from the trivial base case

$$G(\tau_{N-1}) = G(\tau_{N-1:}) = \frac{\sum_i^{m_{N-1}} r_{N-1}^i}{m_{N-1}},$$

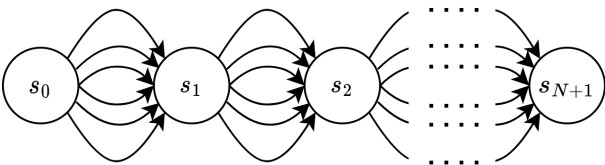

Figure 4: *N*-Step Chain MDP: The MDP is initialized in $s_0$ and terminates in $s_{N+1}$, with actions always leading to the next state in the chain. In this MDP there are $b^N$ possible trajectories.

where the inductive step is:

$$
\begin{aligned}
G(\tau_{t:}) = \frac{\sum_{i=1}^{\Pi_{j=t}^{N-1} m_j} \sum_{k=t}^{N-1} r_k^i}{\Pi_{j=t}^{N-1} m_j} &= \frac{\sum_{i=1}^{\Pi_{j=t}^{N-1} m_j} \left( r_t^i + \sum_{k=t+1}^{N-1} r_k^i \right)}{m_t \Pi_{j=t+1}^{N-1} m_j} \\
&= \frac{\sum_{i=1}^{\Pi_{j=t}^{N-1} m_j} r_t^i}{m_t \Pi_{j=t+1}^{N-1} m_j} + \frac{\sum_{i=1}^{\Pi_{j=t}^{N-1} m_j} \sum_{k=t+1}^{N-1} r_k^i}{m_t \Pi_{j=t+1}^{N-1} m_j} \\
&= \frac{\sum_{i=1}^{m_t} r_t^i}{m_t} + \frac{m_t G(\tau_{t+1:})}{m_t} \qquad (*) \\
&= \frac{\sum_{i=1}^{m_t} r_t^i + G(\tau_{t+1:})}{m_t}
\end{aligned}
$$

Part (B)(3) follows by repeatedly unrolling the last term in equation $(*)$ above. $\square$

As a straightforward consequence of the above Lemma, we have:

**Proposition 2** (Coverage of Trajectory Space). *In a deterministic $N$-step Chain MDP with branching factor $b$, it is possible for $N$-step bootstrapping to (implicitly) capture full "coverage" of the size $b^N$ trajectory space with only $b$ empirical trajectories—an exponential increase in coverage relative to no bootstrapping. Coverage here refers to the percentage of unique trajectories present in the dataset, where we consider two trajectories with the same actions taken in every state as equivalent.*

*Proof.* This follows directly from Lemma 1(B)(1)-(2) if every possible 1-step subtrajectory $\{\tau_j = (s_j, a_j, r_j, s_{j+1})\}$ appears exactly once in the $b$ empirical trajectories. $\square$

We can additionally make the following related statement about the sample complexity of valuing each trajectory in the Chain MDP when rewards are stochastic (note that in the Chain MDP, a trajectory is equivalent to a policy, so that the following proposition is a finite sample complexity bound on "every policy" policy valuation in the Chain MDP).

**Proposition 2a** (Sample Complexity of Policy Valuation in Stochastic Case). *Consider an $N$-Step Chain MDP with branching factor $b$, whose rewards at each action are bounded random variables with $R(s, a) \in [0, 1]$.*

(A) *Suppose we have $n$ samples $\{\tau^{(i)}\}_{i=1\ldots n}$ of each length $N$ trajectory. There are $b^N$ such trajectories, providing a total of $m = Nb^N n$ samples of length 1 subtrajectories. Without bootstrapping, if*

$$
n \geq \frac{N^2}{2\epsilon^2} \log \frac{2b^N}{\delta}, \text{ or equivalently, } m \geq \frac{N^3 b^N}{2\epsilon^2} \log \frac{2b^N}{\delta},
$$

*then, with probability at least $1 - \delta$, we have:*

$$
max_\tau \left( \frac{\sum_i G(\tau^{(i)})}{n} - \mathbb{E}\left[ G(\tau) \right] \right) \leq \epsilon.
$$

*(B) Suppose we have at least $\ell$ samples of each length 1 subtrajectory $\tau_j = (s_j, a_j, r_j, s_{i+1})$. There are $bN$ such subtrajectories, providing a total of $m = bN\ell$. samples of length 1 subtrajectories. Using $N$-Step bootstrapping as described in Lemma 1, if*

$$\ell \geq \frac{N^2}{2\epsilon^2} \log \frac{2bN}{\delta}, \text{ or equivalently, } m \geq \frac{N^3 b}{2\epsilon^2} \log \frac{2bN}{\delta},$$

*then, with probability at least $1 - \delta$, we have:*

$$max_\tau \left( \frac{\sum_i G(\tau^{(i)})}{n} - \mathbb{E}\left[G(\tau)\right] \right) \leq \epsilon.$$

*Therefore, in an $N$-Step Chain MDP with $b > 1$, plain RvS requires $b^{N-1}$ times as many (i.e., exponentially more) samples to obtain the same precision as $N$-step bootstrapping.*

*Proof.*

(A) This follows from Hoeffding's Inequality by noting that $G(\tau) \in [0, N]$ and taking a union bound over the $b^N$ trajectories.

(B) This follows from Hoeffding's Inequality using $\epsilon' = \epsilon/N$, so that the trajectory error (a sum of $N$ 1-step subtrajectories) is bounded by $\epsilon$, and taking a union bound over the $bN$ 1-step subtrajectories.

$\square$

The Chain MDP used in Proposition 2 reveals a general construction to prove necessity, as follows.

**Proposition 1** (Necessity). *For any positive integer $n$, there exists an MDP $\mathcal{M}$ and data generating policy $\pi_e$ for which $n$-step bootstrapping is strictly necessary to generate dataset $\mathcal{D}$ containing $(s, a, g^*)$, where $g^*$ is the maximum reward achievable for a trajectory starting in $(s, a)$.*

*Proof.* For the general case, consider an $n + 1$ state Chain MDP with a single optimal trajectory, where $\mathcal{D}$ is generated by a combination of $n$ non-Markovian policies that each contain a single, unique length 1 segment of the optimal trajectory. A single step of SUPERB bootstrapping can add at most 1 new length 1 segment to any particular (augmented) return label, so that $n$ steps of SUPERB bootstrapping are necessary to compose all $n$ length 1 segments from the optimal policy. $\square$

## B  IMPLEMENTATION DETAILS

Code for our implementation and experiments is available at **[GITHUB LINK TO BE ADDED]**. We implemented our experiments on top of the code for RvS (Emmons et al., 2021) found at `https://github.com/scottemmons/rvs`.

### B.1  RETURN MODEL

To model the distribution of returns we use an ensembled Quantile Regression Network (QRN), as proposed by Dabney et al. (2018), which maps states to a value distribution represented by 20 quantiles. The quantile regression is trained using the Huber quantile loss proposed by Dabney et al. (2018) with $k = 1$, and is optimized for 5 epochs (Antmaze) or 10 epochs (Gym) using AdamW (Loshchilov & Hutter, 2019) using a batch size of 1024, and a constant learning rate of 1e-3 for AntMaze and 3e-4 for the Gym tasks.

Our QRN is an ensemble of 5 feedforward neural networks, each with 3 hidden layers of 512 neurons and ReLU activations. Both the inputs to the networks, and the output targets are normalized. For AntMaze experiments, we apply value clipping to clip target return values to their known feasible range (after reward transformation) of $[-1/(1 - \gamma), 0]$, where $\gamma = 0.997$. In each bootstrapping step, a new randomly initialized QRN is trained on the current augmented dataset.

To form augmented labels, we first use the QRN to propose return labels for all trajectory suffixes in the dataset. This is done by calling each member of the QRN ensemble on the first observation in the trajectory suffix, taking the mean (Antmaze) or minimum (Gym) across the ensemble, and average the top 5 quantiles of the result.

Proposed return labels in hand, we then apply the following backward induction procedure to relabel all return labels in $\mathcal{D}$:

```python
def qr_augmented_return_labels(traj, proposed_labels, discount_factor):
    rewards = traj.rewards
    returns = []
    ret = 0
    traj_len = len(rewards)

    for i in reversed(range(traj_len)):
        ret *= discount_factor
        ret += (float(rewards[i]))
        if (i + 1 < traj_len):
            ret = max(ret, float(rewards[i]) +\
                    discount_factor * proposed_labels[i + 1])
        returns.append(ret)
    returns = list(reversed(returns))
    return returns
```

On each iteration of SUPERB, we keep only the most recently generated augmented labels, and discard both the QRN for that step and the data it was trained on. After the final step, we train one more QRN to use as the value function for the purpose of forming reward targets for the RvS policy, described next.

## B.2 RvS POLICY

For simplicity, we base our policy learning heavily off of the hyperparameters discussed in Emmons et al. (2021). As in Emmons et al. (2021), our policy consists of a simple feedforward neural network with two hidden layers with width equal to 1024. We optimized our policy network using AdamW (Loshchilov & Hutter, 2019), applying weight decay of `1e-2`, a learning rate of `1e-3`, and a batch size of 2048. The learning rate is annealed using a cosine learning rate schedule Loshchilov & Hutter (2016).

In order to support conditioning on higher values of return, we implemented input normalization for the policy network, where inputs are normalized by the mean and standard deviation of the inputs from the dataset. Due to this normalization, policy training could be completed in fewer epochs and we decreased the number of training epochs from 2000 to 400 for D4RL-gym and trained for only 100 epochs on AntMaze.

As discussed in section 3.5, we dynamically choose a return target $g^*$ by using our learned return model (for conditioning we use only the top quantile, rather than take the mean of the top 5, as done for augmentation). However, we found that we can additionally improve performance by increasing the target by some $\Delta$ to just above the value predicted by $V_\phi(G \mid s)$. We tune $\Delta$ separately for each environment and number of iterations, as the return distribution varies with these variables. Table 3 shows that we can still obtain good performance using a fixed $\Delta$ across all tested environments.

| Task | RvS (Ours) | SB (1) | SB (2) | SB (3) | RvS (Ours) - Fixed Δ | SB (1) - Fixed Δ | SB (2) - Fixed Δ | SB (3) - Fixed Δ |
|---|---|---|---|---|---|---|---|---|
| antmaze-umaze-v2 | 84.2 | **85.6** | 85.4 | 83.8 | 79.2 | 8.2 | 0.0 | 0.0 |
| antmaze-umaze-diverse-v2 | 77.7 | 76.5 | 76.9 | **81.2** | 76.9 | 76.5 | 76.2 | 75.4 |
| antmaze-medium-play-v2 | 2.7 | 58.1 | **76.5** | 66.5 | 0.0 | 49.2 | 75.0 | 61.5 |
| antmaze-medium-diverse-v2 | 2.7 | 55.8 | **78.1** | 66.2 | 1.5 | 54.6 | 69.6 | 48.1 |
| antmaze-large-play-v2 | 0.8 | 16.2 | 25.8 | **26.5** | 0.0 | 15.8 | 21.5 | 24.2 |
| antmaze-large-diverse-v2 | 6.9 | 24.2 | **36.9** | 19.6 | 4.2 | 18.1 | **36.9** | 19.6 |
| halfcheetah-random-v2 | 2.2 | 2.2 | 2.2 | **2.3** | 2.2 | 2.2 | 2.2 | 2.2 |
| hopper-random-v2 | 5.8 | 7.9 | 14.3 | **23.4** | 3.6 | 6.8 | 7.6 | 21.9 |
| walker2d-random-v2 | 6.0 | **6.3** | 5.6 | 5.6 | 5.8 | 6.1 | 5.6 | 5.5 |
| halfcheetah-medium-replay-v2 | 39.1 | 39.1 | 39.1 | **39.7** | 38.3 | 38.7 | 38.2 | 39.5 |
| hopper-medium-replay-v2 | 77.9 | 90.6 | 90.3 | **91.0** | 76.3 | 90.2 | 88.8 | 89.7 |
| walker2d-medium-replay-v2 | 50.5 | 56.5 | **66.2** | 59.9 | 50.5 | 56.5 | 63.0 | 59.9 |
| halfcheetah-medium-v2 | 42.3 | 42.3 | 42.1 | **42.4** | 42.2 | 42.1 | 42.0 | **42.4** |
| hopper-medium-v2 | **58.9** | 57.6 | 56.8 | 56.8 | 57.1 | 56.2 | 56.4 | 54.9 |
| walker2d-medium-v2 | 72.4 | 73.5 | **74.7** | 74.2 | 72.4 | 72.1 | 73.4 | 73.8 |
| halfcheetah-medium-expert-v2 | **92.7** | 92.5 | 92.5 | **92.7** | 92.0 | 92.3 | 92.4 | 92.3 |
| hopper-medium-expert-v2 | 109.8 | 109.9 | **110.1** | 108.5 | 108.4 | 109.5 | 109.8 | 108.5 |
| walker2d-medium-expert-v2 | 106.9 | **108.0** | 107.9 | 107.6 | 106.7 | **108.0** | 107.2 | 106.4 |

Table 3: Results when using the best fixed Δ across all D4RL environments. Although there is some advantage to tuning Δ for each environment, we can obtain good performance using a fixed Δ.

| Task | CQL | IQL | BC | % BC | DT | RvS-R | RvS (Ours) | SB (1) | SB (2) | SB (3) |
|---|---|---|---|---|---|---|---|---|---|---|
| antmaze-umaze-v2 | 44.8 | 83.7 | 54.6 | 60.0 | 65.6 | 64.4 | 84.2 | **85.6** | 85.4 | 83.8 |
| antmaze-umaze-diverse-v2 | 23.4 | 66.7 | 45.6 | 46.5 | 51.2 | 70.1 | 77.7 | 76.5 | 76.9 | **81.2** |
| antmaze-medium-play-v2 | 0.0 | 74.7 | 0.0 | 42.1 | 1.0 | 4.5 | 2.7 | 58.1 | **76.5** | 66.5 |
| antmaze-medium-diverse-v2 | 0.0 | 71.0 | 0.0 | 37.2 | 0.6 | 7.7 | 2.7 | 55.8 | **78.1** | 66.2 |
| antmaze-large-play-v2 | 0.0 | **44.3** | 0.0 | 28.0 | 0.0 | 3.5 | 0.8 | 16.2 | 25.8 | 26.5 |
| antmaze-large-diverse-v2 | 0.0 | **49.0** | 0.0 | 34.3 | 0.2 | 3.7 | 6.9 | 24.2 | 36.9 | 19.6 |
| halfcheetah-random-v2 | **18.6** | - | 2.3 | 2.0 | 2.2 | 3.9 | 2.2 | 2.2 | 2.2 | 2.3 |
| hopper-random-v2 | 9.3 | - | 4.8 | 4.1 | 7.5 | 7.7 | 5.8 | 7.9 | 14.3 | **23.4** |
| walker2d-random-v2 | 2.5 | - | 1.7 | 1.7 | 2.0 | -0.2 | 6.0 | **6.3** | 5.6 | 5.6 |
| halfcheetah-medium-replay-v2 | **47.3** | 44.2 | 36.6 | 40.6 | 36.6 | 38.0 | 39.1 | 39.1 | 39.1 | 39.7 |
| hopper-medium-replay-v2 | **97.8** | 94.7 | 18.1 | 75.9 | 82.7 | 73.5 | 77.9 | 90.6 | 90.3 | 91.0 |
| walker2d-medium-replay-v2 | **86.1** | 73.9 | 26.0 | 62.5 | 66.6 | 60.6 | 50.5 | 56.5 | 66.2 | 59.9 |
| halfcheetah-medium-v2 | **49.1** | 47.4 | 42.6 | 42.5 | 42.6 | 41.6 | 42.3 | 42.3 | 42.1 | 42.4 |
| hopper-medium-v2 | 64.6 | 66.3 | 52.9 | 56.9 | **67.6** | 60.2 | 58.9 | 57.6 | 56.8 | 56.8 |
| walker2d-medium-v2 | **82.9** | 78.3 | 75.3 | 75.0 | 74.0 | 71.7 | 72.4 | 73.5 | 74.7 | 74.2 |
| halfcheetah-medium-expert-v2 | 85.8 | 86.7 | 55.2 | **92.9** | 86.8 | 92.2 | 92.7 | 92.5 | 92.5 | 92.7 |
| hopper-medium-expert-v2 | 102.0 | 91.5 | 52.5 | **110.9** | 107.6 | 101.7 | 109.8 | 109.9 | 110.1 | 108.5 |
| walker2d-medium-expert-v2 | 109.5 | **109.6** | 107.5 | 109.0 | 108.1 | 106.0 | 106.9 | 108.0 | 107.9 | 107.6 |

Table 4: We evaluated our method with 1-3 bootstrap iterations, denoted SB(1), SB(2), and SB(3) on the AntMaze and Gym tasks from D4RL (Fu et al., 2020). Across the AntMaze tasks, which require stitching together behaviors from several suboptimal trajectories in order to perform well, SUPERB provides a strong performance improvement over the baseline RvS agents that don't use bootstrapping, outperforming the prior state-of-the-art in 3 of the 6 mazes. The benefits of SUPERB on the Gym tasks are more modest, though we observe improvements over RvS on a few of the suboptimal datasets, particularly those where the dataset was generated by a non-Markovian combination of policies. Generally, most of the performance benefit seems to be captured by a single iteration of SUPERB. * **Bold** denotes the best overall method. Results are averaged over 5 seeds.

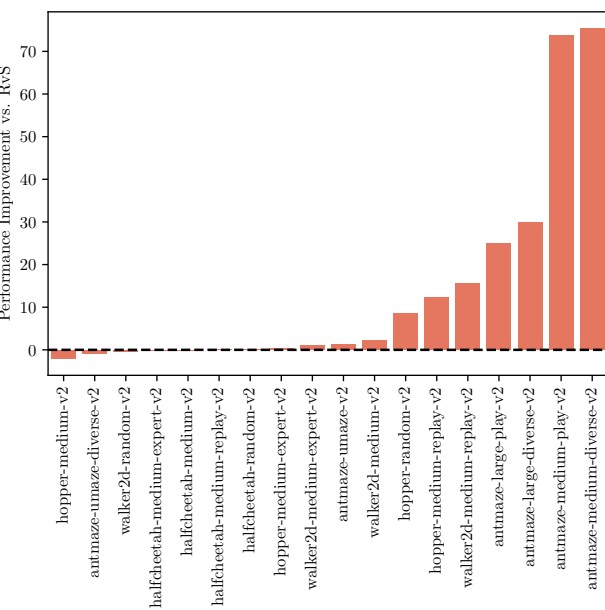

Figure 5: SUPERB gives a substantial boost in performance compared to RvS (ours) across most tasks. This includes both D4RL AntMaze and Gym tasks.

