# OpenReview forum: "Return Augmentation gives Supervised RL Temporal Compositionality"
_ICLR.cc/2023/Conference — Submitted to ICLR 2023_

### Official Review · Reviewer_FZYT · 2022-10-23

**Confidence:** 3
**Correctness:** 4
**Technical Novelty And Significance:** 2
**Empirical Novelty And Significance:** 2
**Recommendation:** 6

**Clarity, Quality, Novelty And Reproducibility:**


The description of their experiment looks clear to me and their result sounds reasonable. In terms of novelty, I would argue that this work is an incremental improvement over the existing RvS method as opposed to a fundamental change in ways of thinking or in the SOTA numbers. Even though there are no clear downsides over RvS, in order for this method to really outperform the baseline, one needs domain knowledge about the task to tune the number of bootstrap iterations.


**Strength And Weaknesses:**


The paper has a clear motivation and diagnosis for the problem they are trying to tackle, a sound logic for their proposed solution (SuperB) accompanied by a proof. They showed that their proposed method improves over the baseline RvS method (the one without privileged goal info) by a significant margin.

The main weakness is where else this approach can be used. As shown in Table 1, it does not result in SOTA performance unless it is clear from the dataset and problem setup that an agent will benefit from stitching together multiple trajectories.  It is also assumed that different trajectories will visit similar states, allowing the bootstrapping to be effective. SuperB is designed specifically for those assumptions and works well on AntMaze medium and umaze. It is less clear why the method works less well on antmaze large compared to IQL.
The authors are aware of the issues of their proposed method and as mentioned in 3.4, “there is no universally optimal number of iterations”, meaning that one needs to either use domain knowledge (like in RvS-G) or do large scale hyperparameter sweeping to find the optimal setup.
Nevertheless, the method looks like a strict improvement over the baseline RvS on the D4RL dataset, and assuming reasonably that the state representation is reasonably good, there does not seem to be a strong downside to swapping out RvS rewards for the SuperB rewards.



**Summary Of The Paper:**

In this paper, the authors observed that RvS, their main baseline method, does not do well on Antmaze when not given privileged information such as the goal location. They hypothesized that by modifying the reward to do bootstrapping, similar to other methods based on temporal difference learning(e.g. CQL, BCQ), that will allow the RvS methods to do better.
They observed no significant benefit on halfcheetah, hopper, and walker2d tasks (apart from the random data split which benefits more from trajectory stitching), but they outperformed the previous SOTA in 3 of the 6 mazes in AntMaze.


**Summary Of The Review:**

This paper proposes SuperB, an incremental improvement over RvS where they introduced bootstrapping to the target reward. The method is shown to work well where it is known that the offline data contains multiple imperfect trajectories that can be combined together to form a better one. I recommended a rating of 6 due to their sound experimental results and due to the limited novelty, but also due to the paper’s thorough discussion of the method’s limitations and future works.

---

> ### Author Response · Authors · 2022-11-17
> **Response to reviewer FZYT**
>
> We would like to thank the reviewer for their insightful comments and suggestions.
>
> > The main weakness is where else this approach can be used.
>
> We disagree. Our goal is to improve the state of supervised RL by endowing it temporal compositionality via data augmentation. While this has the largest effect on environments like AntMaze, which explicitly require stitching behaviors together to get strong performance, SuperB does not significantly hurt performance on other tasks. In fact, on the Gym tasks, it also gives a net-positive performance gain. We updated the appendix with Figure 5 (Appendix B) to illustrate the performance benefits that SuperB achieves across all of our included environments. Finally, we expect that as dataset complexity increases (e.g., due to the higher branching factor, multi-objective nature, and mixture of agents in the real world causing the data-collection policy to be non-Markovian), realistic datasets will look more like antmaze and less like the Gym tasks.
>
> > The authors are aware of the issues of their proposed method and as mentioned in 3.4, “there is no universally optimal number of iterations”, meaning that one needs to either use domain knowledge (like in RvS-G) or do large scale hyperparameter sweeping to find the optimal setup.
>
> Please see Point 3 of our general response to reviewers.
>
> > The description of their experiment looks clear to me and their result sounds reasonable. In terms of novelty, I would argue that this work is an incremental improvement over the existing RvS method as opposed to a fundamental change in ways of thinking or in the SOTA numbers. Even though there are no clear downsides over RvS, in order for this method to really outperform the baseline, one needs domain knowledge about the task to tune the number of bootstrap iterations.
>
> We approximately agree, though we view the value of the paper as: 1) proposing an improvement over RvS; 2) revealing a fundamental insight that RvS and value-based methods are on two different ends of the temporal-compositionality spectrum and SuperB provides a unifying approach that can allows an agent to land somewhere in between to get better performance (see Point 2 of our general response to reviewers).

---

> > ### Comment · Reviewer_FZYT · 2022-12-04
> > **I decide to keep my original score**
> >
> > I agree with reviewer RPpv that the theoretical contribution is somewhat overstated and having the best of both worlds would result in a better paper. I still think that the method is sensitive to the choice of the number of bootstrap iterations, and hiding SB(1) and SB(3) in table 1 of the later revision is not helpful for the reader. It looks like when stitching together trajectories is not required by the environment, the model is not sensitive to the number of steps, but as reviewer JPnx pointed out, in antmaze, the result varies quite a bit for 1-3 iterations.
> >
> > I appreciate the authors' response and further clarifications. I look forward to follow up works in the future.

---

### Official Review · Reviewer_RPpv · 2022-10-24

**Confidence:** 4
**Correctness:** 3
**Technical Novelty And Significance:** 3
**Empirical Novelty And Significance:** 2
**Recommendation:** 5

**Clarity, Quality, Novelty And Reproducibility:**

The paper is very clearly written, and the results are correct and somewhat interesting. The baseline results mostly come from published results, but the authors should release their code to enable reproducibility of their own results. The proposed SuperB approach is surprisingly simple, especially as compared to current value-based offline RL methods, which require complex combinations of Q-learning-or-Actor-Critic-style approaches with distributional reasoning. To me, this is the biggest selling point of the paper: the simplicity of the approach makes it easy to understand, implement, and potentially build upon.

I have multiple concerns about the paper, which make me be mostly on the fence about recommending acceptance. I hope the authors take the time to engage in the discussion and improve their manuscript during the discussion period.

My primary concern regards the motivation for the proposed solution. The authors mention that the main advantages of RvS are "simple training objective, robustness to hyperparameters, and strong performance". However, they also state that value-based RL tends to have higher performance (as demonstrated also in the expeirments of Section 3), and it is unclear in practice whether SuperB in particular is robust to hyperparameters (more on that later). With this, I wonder exactly what the need for another RvS method is, especially as the shortcoming that this paper addresses (RvS's inability to stitch suboptimal trajectories into optimal trajectories) is already handled "for free" by value-based methods. Since we are not gaining improved performance nor (potentially) robustness to hyperparameters, then what is the advantage of RvS with respect to value-based methods? Or, if there are none so far, what are potential advantages in the future of methods that follow a line of work starting from SuperB? Is it only the method's simplicity? And if so, at what cost does it come? Is there a potential path towards methods with the performance levels of value-based offline RL that maintains the simplicity of SuperB?

I also think perhaps the theoretical results are somewhat overstated.
- The authors explicitly state in Section 2.5 that they "show formally how SuperB is necessary to learn optimal policies for some sets of offline RL problems." Similar language is used throughout the paper when referring to Proposition 1. However, what proposition 1 shows is that n-step bootstrapping is necessary for constructing a data set that contains the optimal returns given a state-action pair. This is a substantially weaker result. First, SuperB is not the only conceivable way to incorporate the n-step returns; indeed, the authors themselves state that they made several design choices along the way, which necessarily implies that there are alternatives. Second, not containing the optimal return in the data does not imply that it is not possible to learn optimal policies. Broadly speaking, there are surely other algorithms that could achieve the "temporal compositionality" that SuperB enables (trivially, as the authors mention, value-based methods satisfy that criterion). I think what the authors most likely mean is that temporal compositionality (generally speaking) is needed for RvS learners to achieve optimal performance (in the context of non-Markovian policies, which is clearly stated). This seems to be true almost by definition. That's not to say that the result isn't useful, just that it should be qualified accordingly throughout the paper.
- The second theoretical results seems a bit less useful. First, the result seems to not require a lot of insight: if trajectories overlap, stitching them together leads to more possible trajectories. But more importantly, the simplified chain MDP used to develop this result restricts the amount of insight that we gain from the result. Sure, in the limit, we might obtain an exponential advantage. But, as the authors mention, in the opposite end of the spectrum, we might get no advantage if there is no overlap between trajectories. How are we to interpret what exists in-between? A far more interesting result would consider a more general MDP, and provide results in terms of some intrinsic property of the MDP and data-generating policy to capture some measure of trajectory overlap, and state the gains of trajectory stitching in this setting. Is such a result feasible, or are we doomed to only know what happens at the extreme ends of the space of MDPs?

In terms of the experiments, I have one subtantial concern: how is \Delta, mentioned in the final sentence of the appendices, tuned? The authors do not mention this anywhere, and I think it is critical towards assessing the fairness of the empirical evaluation. How does the authors' hyperparameter tuning compare to the baselines'? Are the authors evaluating the resulting policy online in the environment or using some form of offline performance estimation? If it is the former, I encourage the authors to look at e.g. [1]. Moreover, as the authors stated that robustness to hyperparameters is one of the key advantages of RvS, I would like to see some discussion of how the need to tune \Delta relates to the claimed robustness. Say, if we chose not to tune it, but to set it manually, how would we go about doing that, and how would that affect the obtained results? Similarly, the results seem to show a substantial sensitivity to the number of iterations (N) hyperparameter. How would one go about choosing N more generally?

Also in the experiments, one ablation that I think would have been nice to include is a bootstrapped version of the value function. The authors made the choice to use Monte Carlo samples to train the value function, and there's some reasonable motivation behind that choice. But, was that choice validated empirically? Could we swap in a Q-learning method, or perhaps even a full-fledged off-line RL method, in place of the simple Monte Carlo value estimate? Would we gain anything from such a combination?

############## Additional feedback ##############

The following points are provided as feedback to hopefully help better shape the submitted manuscript, but did not impact my recommendation in a major way.

I would recommend against the use of the term "temporal compositionality", which is commonly used to talk about combining skills with some higher level policy guiding how skills are composed. While closely related, these two are not the same. The term "stitching" seems more appropriate.

Intro
- The intro is very clearly written and gives a good understanding of where the paper is going

Sec 2.2
- I wonder if the reason BCQ and CQL beat BC is truly related to the ability to compose behaviors. BC doesn't train a sequence model (usually, at least), and thus is also able to compose trajectories if they intersect in the data set. Of course, it will just learn to replicate the most commonly observed a given s, so if that corresponds to a poor behavior, then that will be the chosen a. But, this seems to be an issue due to ignoring the reward during training, and not necessarily of compositionality of trajectories.

Sec 2.4
- How is the original data augmented with returns? It would seem that s,a pairs from earlier in the trajectory would have higher returns. Or are these somehow normalized?

Sec 3
- Some results don't quite seem to validate some of the claims in the paper. For example, CQL is worse than all RvS methods on AntMaze, which is supposed to require temporal compositionality.
- What is the point of the reward transformation ablation? It's not really a design choice of SuperB, and it already follows standard practice from prior work.
- Could the authors better clarify exactly how V is used to sample trajectories for evaluation, and what the ablated version in Table 2 does instead? The former should be included in Section 2, as it is an important part of the algorithm itself, which wasn't mentioned until Section 3.5, and only somewhat clarified in the appendices.

Sec 4
- [2] may be a relevant cite for the final paragraph of Sec 4, though it certainly addresses a different problem setting than SuperB

Typos/style/grammar/layout
- Sec 3.5, "Dynamic target return selection": necessarily need -> need
- Appendix B.2: neurial -> neural

[1] Le Paine et al. "Hyperparameter Selection for Offline Reinforcement Learning". 2020

[2] Yu et al. "Conservative Data Sharing for Multi-Task Offline Reinforcement Learning". 2021


**Strength And Weaknesses:**

########### Strengths ###########
1. The paper is clearly written and easy to follow. The motivation is specified clearly from the beginning and repeated/clarified throughout the paper
2. The proposed SuperB approach is simple conceptually, and is a reasonable solution to handle the problem of trajectory stitching from offline data, as demonstrated theoretically and empirically
3. The idea of using a return model to stitch trajectories, instead of attempting to explicitly find trajectories in the data that could be stitched, is clever

########### Weaknesses ###########
1. The theoretical results might be somewhat overstated, and they appear to fall out by construction. They do provide some grounding to validate the proposed solution, so I do think they should be kept in the paper, if perhaps rephrased and downplayed
2. The resulting approach is not particularly strong compared to baselines, especially value-based approaches. While state-of-the-art performance shouldn't be the focus of publications, it is important to discuss and analyze where these ideas might take us in the future if pursued further
3. Some details about the experimental setting are missing and would be necessary for ensuring a fair comparison


**Summary Of The Paper:**

The submission introduces SuperB, a new approach for offline RL following the RL via Supervised Learning (RvS) framework. SuperB addresses a shortcoming of RvS methods, which fail to stitch together trajectories that were executed as parts of non-optimal trajectories in the data, but which would result in an optimal (or high-performing) trajectory if combined into a single trajectory. The approach is simple conceptually: train a return model to estimate the distribution of returns from any given state, and use that to compute potentially higher returns for states in the data set that would be attained if a different trajectory had been followed thereafter, thereby augmenting the data with better trajectories. The authors show theoretically a necessity result (that n-steps returns are necessary for the data set to contain the optimal return if the data-generating policy is Markovian) and a coverage result (that n-step returns enable generating up to exponentially many unseen trajectories). Empirically, SuperB performs competitively to value-based RL and outperforms RvS methods in settings where the data-generating policy is non-Markovian.


**Summary Of The Review:**

Overall, I think this submission is a good contribution and I'm leaning towards recommending its acceptance. My main concerns relate to the motivation for the proposed approach, the fairness of the experimental evaluation, and the somewhat overstated theoretical results. I would like to see the authors' response to these and would be glad to revise my recommendation accordingly.

############# Update after rebuttal #############

I am updating my score from 6 (marginally above threshold) to 5 (marginally below threshold) per the discussion with the authors below.

---

> ### Author Response · Authors · 2022-11-17
> **Response to reviewer RPpv (1/3)**
>
> We would like to thank the reviewer for their insightful comments and suggestions. We agree that SuperB strikes a compelling balance between simplicity and performance. You identified the main concerns as “motivation for the proposed approach, the fairness of the experimental evaluation, and the somewhat overstated theoretical results”. We will address these in turn, followed by our responses to other miscellaneous comments.
>
> **Motivation for the proposed approach**
>
> > Since we are not gaining improved performance nor (potentially) robustness to hyperparameters, then what is the advantage of RvS with respect to value-based methods? Or, if there are none so far, what are potential advantages in the future of methods that follow a line of work starting from SuperB? Is it only the method's simplicity? And if so, at what cost does it come? Is there a potential path towards methods with the performance levels of value-based offline RL that maintains the simplicity of SuperB?
> > The resulting approach is not particularly strong compared to baselines, especially value-based approaches. While state-of-the-art performance shouldn't be the focus of publications, it is important to discuss and analyze where these ideas might take us in the future if pursued further
>
> Please see Point 2 of our general response to all reviewers.
>
> **Fairness of the experimental evaluation**
>
> > Some details about the experimental setting are missing and would be necessary for ensuring a fair comparison
>
> Please see Point 1 of our general response to all reviewers. We have updated the paper in section 2, 3, and in the appendix to improve reproducibility. We also now include code in our supplemental materials and commit to releasing an open-source version compatible with several popular RvS methods upon publication.
>
> We note that our primary empirical objective was to improve upon RvS, and to ensure fair comparison, we re-ran RvS-R with our changes (RvS (Ours) in Table 1).
>
>
> > In terms of the experiments, I have one subtantial concern: how is $\Delta$, mentioned in the final sentence of the appendices, tuned? The authors do not mention this anywhere, and I think it is critical towards assessing the fairness of the empirical evaluation. How does the authors' hyperparameter tuning compare to the baselines'? Are the authors evaluating the resulting policy online in the environment or using some form of offline performance estimation? If it is the former, I encourage the authors to look at e.g. [1].
>
> We recognize that tuning hyperparameters online is often infeasible in the offline-RL setting and we appreciate the reference. However, the common practice in all of our baselines and in offline RL in general is to tune hyperparameters at least somewhat online (Decision Transformer [1] uses different return targets (and even transformer context length) per-game; RvS does the same and mentions that offline tuning with validation loss is not a viable approach; CQL uses a different $\alpha$ per-task; IQL mentions doing a sweep over their expectile parameter online). It is our opinion that tuning hyperparameters (particularly the return target) offline is orthogonal to our proposed research direction - and that our current approach does not stray from the precedent set by these prior works.
>
> Regarding $\Delta$, we tune $\Delta$ per-task online in the same way that RvS/DT tune their return targets. We also provide a new experiment in Figure 5 (Appendix B) which shows that when $\Delta$ is fixed between tasks, performance generally (with the exception of one task) is quite similar.
>
> > Also in the experiments, one ablation that I think would have been nice to include is a bootstrapped version of the value function. The authors made the choice to use Monte Carlo samples to train the value function, and there's some reasonable motivation behind that choice. But, was that choice validated empirically? Could we swap in a Q-learning method, or perhaps even a full-fledged off-line RL method, in place of the simple Monte Carlo value estimate? Would we gain anything from such a combination?
>
> This is a good question. We suspect that if we used a full-fledged value-based offline RL method (i.e., train a Q-function with CQL or IQL, and then use these Q-values to train an RvS policy) we would likely just end up with a policy that is quite similar to the original offline RL method. This is because the algorithm resembles the way that neural network policies are already used in order to do amortized Q-value maximization in continuous Q-learning methods. There probably isn’t a big benefit to doing it with RvS vs. what is already done since there would not be any diversity in the return-labels that each (s, a) pair gets assigned. This is in contrast to SuperB, where the TD relation is used to generate new sets of possible returns that could be achieved from a specific (s, a) pair.

---

> > ### Author Response · Authors · 2022-11-17
> > **Response to reviewer RPpv (2/3)**
> >
> > **Overstatement of Theoretical Results**
> >
> > We believe we understand your general criticism and have adjusted the language in the paper to incorporate your suggestions re: SuperB being only one possible solution to temporal compositionality. Please let us know if you think further edits are necessary.
> >
> > > The second theoretical results seems a bit less useful. First, the result seems to not require a lot of insight: if trajectories overlap, stitching them together leads to more possible trajectories. But more importantly, the simplified chain MDP used to develop this result restricts the amount of insight that we gain from the result. Sure, in the limit, we might obtain an exponential advantage. But, as the authors mention, in the opposite end of the spectrum, we might get no advantage if there is no overlap between trajectories. How are we to interpret what exists in-between? A far more interesting result would consider a more general MDP, and provide results in terms of some intrinsic property of the MDP and data-generating policy to capture some measure of trajectory overlap, and state the gains of trajectory stitching in this setting. Is such a result feasible, or are we doomed to only know what happens at the extreme ends of the space of MDPs?
> >
> > As noted in the main text, Proposition 2/2a provide an analysis that is highly favorable to bootstrapping, and are intended to show that bootstrapping can provide exponential improvements over plain RvS *even with a Markovian data-generating policy*. For a non-Markovian data-generating policy, there are cases where no amount of samples will be able to match the performance of bootstrapping (e.g., figure 1).
> >
> > While it’s possible that one could construct something in between, which may or may not maintain an exponential gain in sample complexity, we believe it would still require fairly specific assumptions about the MDP and we are not sure it would provide much additional insight. Re: a possible intrinsic property that would be general to all MDPs, we agree this would be more interesting—unfortunately, while we could write down an algorithm to count the total number of trajectories based on where they intersect, it’s not clear to us how to simplify this into a simple summary metric, even for 1-step of bootstrapping.
> >
> > **Miscellaneous**
> >
> > > I would recommend against the use of the term "temporal compositionality", which is commonly used to talk about combining skills with some higher level policy guiding how skills are composed. While closely related, these two are not the same. The term "stitching" seems more appropriate.
> >
> > We debated whether or not this was the right term to use and ultimately decided to use “temporal compositionality” due to its prior use when referring to the same phenomenon in AntMaze in prior offline RL works (e.g., page 8; https://arxiv.org/pdf/2106.02039.pdf)
> >
> > > I wonder if the reason BCQ and CQL beat BC is truly related to the ability to compose behaviors. BC doesn't train a sequence model (usually, at least), and thus is also able to compose trajectories if they intersect in the data set. Of course, it will just learn to replicate the most commonly observed a given s, so if that corresponds to a poor behavior, then that will be the chosen a. But, this seems to be an issue due to ignoring the reward during training, and not necessarily the compositionality of trajectories.
> >
> > We agree with you that BC and RvS methods could potentially compose behaviors via the policy (albeit in a limited form); e.g., if trajectory A has the highest reward from the initial state, then RvS would follow trajectory A initially; but if A intersects with trajectory B, which has a higher return than trajectory A for the remainder, then RvS would switch to trajectory A. We argue that such composition is poor as it is in a sense “lucky”: like BC, it is not intentional / conditioned on the maximal reward. As suggested by Figure 1 and the results in Figure 3, RvS does not, generally speaking, compose behaviors.  We agree that there may exist other ways to drive the RvS policy to stitch together optimal behaviors—e.g., by conditioning it on initially suboptimal returns in order to drive it to relevant points of intersection—but this would be a different method from SuperB.
> >
> > > How is the original data augmented with returns? It would seem that s,a pairs from earlier in the trajectory would have higher returns. Or are these somehow normalized?
> >
> > Indeed, (s, a) pairs from early in a trajectory will have higher returns (at least if no discounting is used). We only normalize based on the mean and std of the offline dataset/return labels. Normalizing based on the amount of time remaining in the trajectory is something that was done in the original RvS paper, but we found that this design choice didn’t positively impact performance.

---

> > > ### Author Response · Authors · 2022-11-17
> > > **Response to reviewer RPpv (3/3)**
> > >
> > > > Some results don't quite seem to validate some of the claims in the paper. For example, CQL is worse than all RvS methods on AntMaze, which is supposed to require temporal compositionality.
> > >
> > > We are also confused by the performance of CQL on the AntMaze tasks and suspect that with different hyperparameters they may be able to work (we use results that are reported in the RvS paper, since the original CQL paper did not use the updated version of the AntMaze environment that we evaluate on). IQL, which we evaluate ourselves on AntMaze-v2, does validate our claims in the paper. We may run our own version of CQL in order to update the table before publication.
> > >
> > > > What is the point of the reward transformation ablation? It's not really a design choice of SuperB, and it already follows standard practice from prior work.
> > >
> > > This is something that no prior RvS method used for AntMaze. We wanted to show that while this reward transformation is necessary to get AntMaze working with RvS (table 2), it isn’t sufficient (table 1; RvS [ours] uses the reward transformation and still performs poorly).
> > >
> > > > Could the authors better clarify exactly how V is used to sample trajectories for evaluation, and what the ablated version in Table 2 does instead? The former should be included in Section 2, as it is an important part of the algorithm itself, which wasn't mentioned until Section 3.5, and only somewhat clarified in the appendices.
> > >
> > > We have added these details to the revised paper. As noted in Appendix B, the sampling procedure takes the mean of the upper 5 (of 20) quantiles of the QRN (we had initially chosen top 1, which was unstable, and we also tried values of 3 and 7, which also worked).
> > > The ablation in Table 2 considers whether we condition the RvS policy on the return specified by a learned value function V (trained on the final dataset, as noted at the end of Appendix B.1), or whether we specify an absolute reward target, as is typical of RvS methods.
> > >
> > > —
> > >
> > > We thank you again for your detailed and thoughtful review and we hope that we have addressed your concerns.

---

> > ### Comment · Reviewer_RPpv · 2022-11-22
> > **Response to authors**
> >
> > I thank the authors for their careful response. I repeat below my main concerns, state how the authors addressed those concerns, and discuss my updated view of the paper.
> > 1. **Overstated theoretical results.** The authors downplayed the strength of their claim surrounding Proposition 1, and I think that improves the quality of the manuscript. The authors further discuss how the result in Proposition 2 might be interesting.
> > 2. **Clarity of the motivation, given lower performance than value-based methods.** The authors clarify that the trade-off between value-based and supervised methods is that the former is good at trajectory stitching and the latter is good at scaling with the number of parameters and avoiding the deadly triad; SuperB is proposed as an in-between method that should get the "best of both worlds". I believe this is a solid motivation. However, the results are focused on validating that SuperB retains most of the trajectory stitching properties of value-based methods, but nothing on how they retain scalability of supervised methods. I would encourage the authors to dive deeper into that second half of the claim in updated experiments in a future revision of this work.
> > 3. **Fairness of evaluation / robustness to hyperparameters.** The manuscript now describes how $\Delta$ was tuned, and the authors contend that online hyperparameter tuning is standard practice. I agree with this argument. However, there is no discussion of how the need to tune $\Delta$ relates to the authors' claim that supervised methods are less sensitive to hyperparameters than value-based methods. Indeed, the only result analyzing this claim was the sensitivity to N, the number of bootstrap iterations; the authors have now removed these results because they didn't quite show robustness. I do not believe that removing sensitivity to N is an appropriate choice, and I would encourage the authors to put that sensitivity analysis back in, or remove the claim of robustness to hyperparameters altogether, as there is now no result in the manuscript to validate that claim.
> >
> > Overall, I believe that unfortunately the authors' response did not improve my perception of the paper, and I am in fact now leaning to recommending the paper for rejection, with encouragement to continue improving the manuscript for a future submission.

---

### Official Review · Reviewer_JPnx · 2022-10-25

**Confidence:** 5
**Clarity, Quality, Novelty And Reproducibility:** Mentioned above.
**Correctness:** 3
**Technical Novelty And Significance:** 3
**Empirical Novelty And Significance:** 3
**Recommendation:** 5

**Strength And Weaknesses:**

Strengths
- The paper deals with an important problem of return-conditioned supervised learning methods, which is that they cannot stitch trajectories to find optimal behavior.
- The proposed return augmentation method does seems to improve performance on various benchmarks.

Weaknesses
- The source code is not provided, which makes the credibility of the work questionable.
- As the authors note, a major advantage of the return-conditioned SL methods is that they do not require value bootstrapping and thus does not fall into the infamous 'deadly triad'. However, adopting the return augmentation with iterative value updates can reintroduce this bootstrapping problem.
- The return conditioning procedure is not explained in detail even though the algorithm's performance depends heavily on it (Table 2). How is the value model V_\phi used to sample "high, but plausible returns for the current episode"? Also, Appendix B.2 notes that the targeted value generated from the value model is increased by some \Delta. How is this \Delta tuned? Overall, how much does each design choice affect the performance?
- The performance of the proposed method is highly sensitive to N, the number of bootstrapping iterations, especially on Antmaze.
- The D4RL benchmark results are averaged over only 3 seeds. Adding at least 1~2 more seeds would be more credible.

Questions
- What does the numbers on Figure 3 (b) mean? Does it refer to the expected value (from V_\phi) ?

############## Post-rebuttal comment ##############

I appreciate the authors for updating the manuscript, but there still are some remaining concerns.

First, the hyperparameter tuning procedure is still somewhat ambiguous. The tuning process (search range, selection method, ...) for \Delta is not included in the paper. Also, the authors mention that "each method has one set of hyperparameters shared across all tasks" on the updated Table 1, but this is wrong since the proposed method tunes \Delta per task.

Second, I agree that the work can be interpreted as unifying value-based methods and supervised-RL, but given the empirical results, I think the proposed method fails to achieve the best of both worlds (the proposed method outperforms value-based baselines on only Antmaze). I encourage the authors to improve the method so that the cons from both approaches are minimized while the pros are maximized.

**Summary Of The Paper:**

The paper proposes a data augmentation method for return-conditioned supervised learning that takes into account temporal compositionality. The proposed method shows higher performance compared to vanilla baselines.

**Summary Of The Review:**

The paper provides a data augmentation method for return-based supervised learning in offline RL. While finding an appropriate data augmentation technique for return-based supervised learning (and in general, offline RL) is important, the proposed method seems to have some drawbacks.

---

> ### Author Response · Authors · 2022-11-17
> **Response to reviewer JPnx**
>
> We would like to thank the reviewer for their insightful comments and suggestions. Please see Point 1 of our general response to reviewers re: source code.
>
> > As the authors note, a major advantage of the return-conditioned SL methods is that they do not require value bootstrapping and thus does not fall into the infamous 'deadly triad'. However, adopting the return augmentation with iterative value updates can reintroduce this bootstrapping problem.
>
> Please see Point 2 of our general response to reviewers.
>
> > The performance of the proposed method is highly sensitive to N, the number of bootstrapping iterations, especially on Antmaze.
>
> Please see Point 3 of our general response to reviewers.
>
> > The return conditioning procedure is not explained in detail even though the algorithm's performance depends heavily on it (Table 2). How is the value model V_\phi used to sample "high, but plausible returns for the current episode"? Also, Appendix B.2 notes that the targeted value generated from the value model is increased by some $\Delta$. How is this $\Delta$ tuned? Overall, how much does each design choice affect the performance?
>
> Please see Appendix B.2 in our updated paper for more details on how return conditioning works. Regarding the $\Delta$ tuning, in our experiments we pick the best performing $\Delta$ for each task/method. This matches the common practice in the RvS literature [1, 2] of tuning the return target and some other hyperparameters using per-game (online data). In order to show that the choice of $\Delta$ is not too important, we added a new “shared $\Delta$” experiment in Table 3 (Appendix B). In all but one environment, a shared $\Delta$ value performs nearly as well as the tuned one.
>
> As noted in Appendix B, to sample high but plausible returns, our current experiments take the mean of the upper 5 (of 20) quantiles of the QRN (we had initially chosen top 1, which was unstable, and we also tried values of 3 and 7, which also worked).
>
> > The D4RL benchmark results are averaged over only 3 seeds. Adding at least 1~2 more seeds would be more credible.
>
> We will update the paper with extra seeds. We are prioritizing the other changes and releasing our responses to the reviewers, so we will update the paper once more with the extra seed results before the end of the week.
>
> > What does the numbers on Figure 3 (b) mean? Does it refer to the expected value (from V_\phi) ?
>
> Thanks for pointing this out to us. The numbers in Figure 3 (b) are discounted return labels for states corresponding to each location in the grid (you can think of this as a value function). To make this plot, we first cluster the augmented (s, a, return) tuples that the RvS agent is trained on according to where the state is in an x-y grid representing the maze. For each pixel/cluster in the grid, we take the maximum over all return labels. We realize that this figure is unclear in our original submission, and we have clarified this in the revised figure caption.
>
> > The paper provides a data augmentation method for return-based supervised learning in offline RL. While finding an appropriate data augmentation technique for return-based supervised learning (and in general, offline RL) is important, the proposed method seems to have some drawbacks.
>
> Thanks! We believe that SuperB is an important step towards improving our understanding of the limitations of supervised RL methods and their connection to other offline RL methods (we can push supervised RL towards other value-based methods by using data augmentation). While no method is without drawbacks, we hope that you would agree we have discussed these drawbacks thoroughly in our paper such that SuperB can contribute meaningfully to progress in advancing the state of supervised RL.
>
>  If you find our answers and changes responsive to your concerns, we would be grateful if you would consider increasing your score.
>
> [1] “RvS: What is Essential for Offline RL via Supervised Learning?” (Emmons et. al. 2021)
>
> [2] “Decision Transformer: Reinforcement Learning via Sequence Modeling” (Chen et. al. 2021)

---

> > ### Author Response · Authors · 2022-11-17
> > **Additional seeds added**
> >
> > We have now updated the experimental results in the paper to include 5 seeds.

---

### Official Review · Reviewer_sHVj · 2022-10-25

**Confidence:** 4
**Correctness:** 4
**Technical Novelty And Significance:** 4
**Empirical Novelty And Significance:** 3
**Recommendation:** 6

**Clarity, Quality, Novelty And Reproducibility:**

I believe the contribution is novel and writing quality is good. As pointed out in "weakness", I think there are rooms for improvements for clarity and reproducibility. I would also suggest open-sourcing the code to improve reproducibility.

**Strength And Weaknesses:**

Strength:
* Very interesting and novel proposal that addresses the critical bottleneck of decision transformers (DT) or RvS of not being able to infer the optimal return by bootstrapping. I am not aware of previous works that successfully combine DT/RvS and TD-bootstrapping.
* The empirical results on D4RL are very impressive, especially on AntMaze, where an optimal policy must stitch together behaviors from several suboptimal demonstrations, from almost zero to reasonable performance.

Weakness
* How quantile regression was perform and how bias was added into sampling (specifically the implementation related to Dabney et al. (quantile regression) and Lee et al. (multi-game decision transformers)) was not clearly described. I suppose these could be very important to the stability of offline TD-learning. I would like to see more details and potentially more sensitivity analysis on these.
* Many details are missing: Hyper-parameters related to the above point are missing. Network architecture is also missing. Although the reader can refer to code from previous work, it's better to make the paper self-contained and improve reproducibility.

**Summary Of The Paper:**

This paper proposed using value bootstrapping to augment the return data for learning decision transformers or RvS (RL via supervised learning) in general. This approach, which performs explicit bootstrapping / stitching, allows mapping optimal actions to optimal return-to-gos that do not appear in the actual experience and thus achieve better performance.

**Summary Of The Review:**

I lean to acceptance. However, I would like to see if the authors can improve on clarity of implementation details, provide more insight from sensitivity / ablation studies, and increase reproducibility.

---

> ### Author Response · Authors · 2022-11-17
> **Response to reviewer sHVj**
>
> We would like to thank the reviewer for their insightful comments and suggestions.
>
> > How quantile regression was perform and how bias was added into sampling (specifically the implementation related to Dabney et al. (quantile regression) and Lee et al. (multi-game decision transformers)) was not clearly described. I suppose these could be very important to the stability of offline TD-learning. I would like to see more details and potentially more sensitivity analysis on these.
>
> We agree that more details would be helpful and we added new writing in our revision. Please see Appendix B.1 for information on the quantile regression and Appendix B.2 for information on how we choose which return to condition the agent on. We are also including the code used to run our experiments.
>
> Both of the networks were based on existing code and follow the past work closely. Although we did test different layer sizes, neither was extensively tuned. The sampling procedure takes the mean of the upper 5 (of 20) quantiles of the QRN (we had initially chosen top 1, which was unstable, and we also tried values of 3 and 7, which also worked).
>
> Regarding other sensitivity analyses, Table 2 includes an ablation for removing the dynamic target return altogether (it reduces performance). We are also including a new experiment where we choose the same $\Delta$ for each environment in Table 3 (Appendix B). As you can see, with the exception of one environment, using a fixed $\Delta$ performs reasonably well.
>
> > Many details are missing: Hyper-parameters related to the above point are missing. Network architecture is also missing. Although the reader can refer to code from previous work, it's better to make the paper self-contained and improve reproducibility.
>
> We have added some additional details to Appendix B on these points. We hope that this combined with our released code will address your concerns on this point.

---

### Author Response · Authors · 2022-11-17
**General Comments to All Reviewers**

We thank all the reviewers for their time and helpful commentary. We appreciate that you found our work “interesting” (sHVj, RPpv), “novel” (sHVj), “clearly motivated” (RPpv, RZYT), “conceptually simple” (RPpv), and “well written” (sHVj, RPpv), and our empirical improvement over the baseline significant (sHVj, JPnx, FZYT). We’ve responded to each reviewer individually, uploaded a revised draft (revisions shown in green), and created the below responses to shared concerns. If you find that our responses address your concerns, we would be grateful if you would consider increasing your score.

> **[Point 1]** Reviewers sHVj, JPnx and RPpv had concerns over reproducibility and lack of source code

We have attached the code as supplementary material and the plan is for the code to be refactored and released open-source upon publication in order to make it simple for the community to use our augmented returns to train a variety of different RvS agents.

In an effort to make the paper self-contained, we have also expanded the Appendix to include additional implementation details about the networks architectures, hyperparameters, and experimental setup.

> **[Point 2]** Reviewers JPnx, RPpv and FZYT had questions/comments regarding “reintroducing [the] bootstrapping problem” of value-based methods and the advantage over value-based methods given that bootstrapping is introduced into RvS.

We have updated Section 2.4 to clarify that, in some sense, SuperB offers a way to unify RvS and value-based methods (cf. TD(λ) and other unifying methods).  At one end of the spectrum, RvS does not use bootstrapping (doesn’t suffer from the deadly triad) but cannot deal with tasks that require stitching. At the other end, Q-learning methods bootstrap many times (depending on how the target network is implemented) and suffer from the deadly triad, but work well on tasks that require stitching. If a task does not require any stitching, we would expect RvS to work well. If a task requires a large amount of stitching, methods like DQN, which bootstrap many times, are likely to work better. But for many realistic problems (including those in D4RL), when a small but non-zero amount of stitching is required, then an algorithm somewhere in between may be better than either extreme (see Point 3 below).

We’d also like to emphasize that our initial goal when pursuing this work was to push forward our understanding and the performance of supervised RL. To this end, the comparison to pure value-based methods is meant to show how, by incorporating some elements of value-based methods into RvS, SuperB brings RvS to within striking distance of SoTA offline-RL methods on D4RL. We believe work on improving supervised RL is important since supervised RL has shown promise in areas where value-based methods struggle. For example, [1] shows in figure 5 that Decision Transformer (an RvS method) scales well with the number of parameters when trained on 46 Atari games simultaneously while CQL loses performance. This suggests that in the future it may be the case that the RvS end of the spectrum becomes relatively more attractive. In particular, we speculate that supervised RL will become increasingly important as a method for making the most out of large, pretrained generative models.

> **[Point 3]** Reviewers JPnx and RPpv noted that the performance of SuperB is sensitive to the number of bootstrap iterations.

We believe we have misrepresented our results in Table 1 such that it makes our method seem more sensitive to hyperparameters than other methods. It is standard practice for Deep RL papers to pick one set of hyperparameters across all tasks (i.e., Rainbow DQN [2] uses the same hyperparameters across all tested Atari 2600 games). However, it is common knowledge that in such experiments, if hyperparameters were to be tuned per-task, performance would dramatically improve (e.g., Figure 6 in [2]). While we do fix a set of hyperparameters across each task for the most part, the decision to include different values of N in the main table gives the impression that SuperB is more sensitive to hyperparameters or relies more on domain knowledge than the baselines. In reality, if the baselines were presented as including different hyperparameters per-task, then they would likely be perceived as equally sensitivity.
In our case, we find that SuperB with 2 bootstrap iterations (SB(2)) performs well across all tested tasks, so specific domain knowledge is not strictly necessary. To improve the clarity and fairness of the comparison in table 1 (each method has one set of hyperparameters shared across all tasks), we have opted to fix N = 2 and remove the other values of N. The original table has been moved to table 4 in the appendix.

[1] “Multi-Game Decision Transformers” (Lee et. al. 2022)

[2] “Rainbow: Combining Improvements in Deep Reinforcement Learning” (Hessel et. al. 2017)

---

### Decision · Program_Chairs · 2023-01-20

**Decision:**

Reject

**Justification For Why Not Higher Score:**

The paper presents an interesting and novel approach, but important remaining limitations are noted by reviewers. The procedure for hyperparameter tuning remains unclear, and authors should remove the claim of robustness to hyperparameters. The method performs relatively lower on some of the MuJoCo tasks, and an error analysis would provide valuable additonal insights.

**Justification For Why Not Lower Score:**

N/A

**Metareview: Summary, Strengths And Weaknesses:**

This paper addresses a problem of data sparsity in offline reinforcement learning, which results in typically low performance in reward ranges that have not been observed in the training data. The authors proposes an approach called SuperB, which augments data by combining intersecting trajectories. Empirical validation is performed on D4RL.

Reviewers found the paper well written, with an important and well-motivated approach that addresses a key limitation of RL as Supervised learning (RvS) approaches. The approach makes "clever" use of a return model and achieves encouraging results, especially on the AntMaze domain.

Initial reviews noted a number of concerns, some of which have been addressed in the rebuttal phase. More clarity was needed on the description of components of the approach (e.g., how quantile regression was applied) and for the paper to be self-contained. Some felt that the provided theoretical results were over-stated. Reviewers raised questions about the sensitivity to hyperparameter N (the number of bootstrap iterations), and raised the need to better understand limitations of the proposed approach compared to value-based approaches (e.g., relatively lower empirical performance on MuJoCo tasks).

Authors replied in detail, updating the paper with clarifications, providing access to the source code, and adding experimental details to the appendix. They addressed concerns about hyperparameter sensitivity by moving a corresponding table to the appendix, and arguing that the table gave a misleading impression.

Some of the reviewers concerns were addressed in the rebuttal, but some of the concerns remained. In particular, the hyperparameter tuning procedure was still deemed ambiguous, and the choice of moving sensitivity analysis of the effects of N was not considered an appropriate choice. Reviewers argued that claims about the robustness of the approach to hyperparameters should be removed given the lack of evidence. Additional questions focused on a more detailed understanding of the claim that the method provided a "best of both worlds" approach between value-based and RvS methods, both  conceptually and empirically. An error analysis of SuperB on an environment where it achieves relatively lower performance has potential to generate valuable insights.